# A Lightweight Authentication and Key Agreement Schemes for IoT Environments

**DOI:** 10.3390/s20185350

**Published:** 2020-09-18

**Authors:** Dae-Hwi Lee, Im-Yeong Lee

**Affiliations:** Department of Computer Science and Engineering, Soonchunhyang University, Asan 31538, Korea; leedh527@sch.ac.kr

**Keywords:** ECQV implicit certificate, CL-PKC, authentication, key agreement

## Abstract

In the Internet of Things (IoT) environment, more types of devices than ever before are connected to the internet to provide IoT services. Smart devices are becoming more intelligent and improving performance, but there are devices with little computing power and low storage capacity. Devices with limited resources will have difficulty applying existing public key cryptography systems to provide security. Therefore, communication protocols for various kinds of participating devices should be applicable in the IoT environment, and these protocols should be lightened for resources-restricted devices. Security is an essential element in the IoT environment, so for secure communication, it is necessary to perform authentication between the communication objects and to generate the session key. In this paper, we propose two kinds of lightweight authentication and key agreement schemes to enable fast and secure authentication among the objects participating in the IoT environment. The first scheme is an authentication and key agreement scheme with limited resource devices that can use the elliptic curve Qu–Vanstone (ECQV) implicit certificate to quickly agree on the session key. The second scheme is also an authentication and key agreement scheme that can be used more securely, but slower than first scheme using certificateless public key cryptography (CL-PKC). In addition, we compare and analyze existing schemes and propose new schemes to improve security requirements that were not satisfactory.

## 1. Introduction

The Internet of Things (IoT) is an environment/technology in which heterogeneous devices connected to the internet provide various services. Data collected by sensors and actuators are processed by smartphones. The number of IoT devices connected to the internet will increase rapidly in the 5G era [1,2]. People, objects, and spaces are becoming increasingly interconnected. Many countries, including Korea, are investing heavily in the field. The first IoT environment was the smart home, in which IoT technology connects household appliances to the internet. The user can remotely control air conditioners or the boiler to adjust the temperature. Many products featuring artificial intelligence are being released [3]. Mass-produced devices are becoming lighter, and smart buildings, factories, and cities are under construction [4]. Previously, devices could not be connected directly to the internet, requiring a gateway. Today, direct connections allow devices (such as smartphones) to interact. Security is of prime concern, particularly authentication and key management; the latter creates the session keys required for secure communication after authentication. Authentication is an important technology that can be applied in the perception layer and transportation layer, which are the basis of the IoT service [5]. However, existing authentication protocols are inadequate in environments featuring multiple devices.

Figure 1 shows a smart factory wherein IoT devices monitor and control the production equipment [6]. Authentication and key negotiation are required to deliver information quickly and securely. Similarly, when data are sent to the manufacturing execution system (MES), authentication and key agreement must be performed by end-to-end communication via a gateway (GW). However, existing public key infrastructure (PKI)-based authentication is too slow in real-time environments.

Here, we developed authentication and key agreement protocols that create secure keys after mutual authentication to allow IoT objects to communicate. The first scheme allows rapid authentication and key agreement using an implicit certificate termed the elliptic curve Qu–Vanstone (ECQV). Implicit certificate is a way to implicitly authenticate the other party by deriving the public key from the certificate. The second scheme is more secure than the first, but slower, and is an authentication and key agreement using the certificateless public key cryptosystem (CL-PKC). Both schemes use identity (ID)-based PKCs; the first scheme features only implicit authentication. The second scheme incorporates signature information into the public user key.

The contributions of this paper can be summarized as follows.

We analyze existing lightweight authentication and key agreement schemes for IoT environments.In an environment where fast communication is required, we propose a scheme that enables rapid mutual authentication and key agreement through ECQV implicit certificates. This scheme provides implicit authentication for public keys (Scheme 1).Although slower than Scheme 1, we propose an efficient authentication and key agreement scheme based on CL-PKC that allows explicit verification of public keys (Scheme 2).

This paper is organized as follows. Section 2 contains more details on implicit certificates and CL-PKCs. Section 3 pertains to the security requirements. Our two schemes and their development are described in Section 4 and Section 5, respectively. Section 6 contains the conclusion.

## 2. Background and Related Work

In this section, we discuss background and related work. First, we examine what type of authentication and key agreement (AKA) is used in the recent IoT environment. Further, we analyze the AKA schemes using public key certificates and examine the ECQV implicit certificate. We also analyze the certificateless-based AKA (CL-AKA) schemes using the certificateless PKC. Finally, we analyze the existing schemes.

### 2.1. Authentication and Key Agreement (AKA)

The IoT requires efficient and secure key management. Many objects are interconnected, and AKA is required for secure communication [7]. Key management protocols are divided into key distributions and key agreements (or key exchanges). During key distribution, a sender requesting communication generates a session key, and a receiver decrypts that key. Key agreement calculates a session key via the exchange of random values; the key is not transmitted directly. In general, the IoT uses key agreement because the risk of secret key exposure falls when sessional keys are generated via communication between two objects. However, additional authentication processes are required; most basic Diffie–Hellman key agreement schemes are vulnerable to man-in-the-middle and masquerade attacks [8], because the key agreement protocol per se does not feature authentication of mutual objects. Thus, an authentication process is added, and key agreement is performed sequentially. In the IoT environment, it must be confirmed that two communicating objects are legitimate users or devices; this is termed authentication. As shown in Figure 2, a mutual authentication protocol using secret information generally requires an intermediary (e.g., a gateway) that manages secret information and assists with authentication; this is termed three-party key exchange [9,10]. Another scheme features mutual authentication via a certificate issued by a certificate authority (CA), as shown Figure 3 [11]. The advantage is that two objects can communicate directly; there is no gateway. If authentication is lacking, it is possible that an attacker can participate in communication. After authentication, a session key is required to transmit/receive secure data. The session key is securely distributed to users/objects authenticated via the AKA protocol. In recent years, studies on performing mutual authentication using blockchain in authentication and key agreement have also been conducted [12,13].

### 2.2. AKA with ECQV Implicit Certificate

A typical AKA protocol features key agreement based on the Diffie–Hellman approach. PKCs resolve the key management problem of symmetric key cryptosystems, encrypting data or performing digital signatures using both a private and public key. However, if public key authentication is lacking, a man-in-the-middle attack is possible, and trust in the public key must be assumed. Currently, PKIs featuring public key certificates signed by a third-party CA are used to ensure key reliability. However, PKIs are complex; generation, distribution, storage, and disposal of public certificates are required, and verification costs are high.

The ECQV scheme issues an implicit certificate as defined by standard efficient cryptography in SECG SEC 4 [14]. Generally, a public key certificate issued to a user includes an identifier, the key, and a digital signature. The user explicitly authenticates the message by verifying the digital signature using the key and identifier. An implicit certificate includes only the identifier and public key recovery data.

The certificate and key are implicitly verified by computing the public key of the user via the identifier and key recovery data. As no public key is included, an implicit certificate is smaller than a public key certificate. The key is derived by elliptic curve cryptography (ECC); the key length is shorter and computation is more rapid compared with other encryption schemes. An implicit certificate is appropriate for a resource-limited IoT environment. Table 1 compares the ECC with the RSA public key and certificate. Table 2 shows the key lengths and certificate sizes by security strength of comparison of ECQV, elliptic curve digital signature algorithm (ECDSA), and RSA.

An ECQV implicit certificate can be used to perform the certificate-based AKA introduced in Section 2.1. A session key is generated via Diffie–Hellman key exchange, and the public key is restored. This reduces both the key length and certificate size (both are large in existing PKIs), and the session key is generated quickly.

### 2.3. AKA with Pairing-Free Certificateless PKC

Shamir was the first to develop an ID-based cryptosystem allowing management of PKI certificates [15]. In an ID-based PKC, the key distribution problem is solved using a known public key (an ID) rather than an existing authorized certificate. At this time, a trusted third party termed a key generation center (KGC) or a private key generator generates and issues a private key for each user ID. However, all ID-based PKCs suffer from a key escrow problem; the KGC can decrypt all ciphertexts and forge signatures because the KGC generates the private keys. In 2003, Al-Riyami et al. [16] developed a CL-PKC to solve both the public key authentication and key escrow problems. In this cryptosystem, the KGC generates only part of the user’s private and public keys, and the user completes the keys. In other words, in a CL-PKC, the key escrow problem is solved because the KGC knows only some of the private key. The CL-PKC cryptosystem allows data encryption, digital signature, and AKA. The latter features an interactive protocol, and two users negotiate a common session key over a network. Al-Riyami et al. were the first to develop certificateless authentication of key matching based on a CL-PKC. However, as pairing is required, the computational efficiency is low.

### 2.4. Analysis of Existing Schemes

#### 2.4.1. Fast AKA Schemes Including ECQV-Based AKA

In certificate-based AKA, an implicit certificate is received after a user is registered by a CA in the form of a certificate. In ECQV-based AKA, authentication is performed by verifying the public key of the implicit certificate. However, ownership of the public key remains unknown. Because ECQV does not verify the integrity of a signature by reference to the digital signature of the public key like PKI, but performs authentication by calculating the public key of the implicit certificate, public key replacement and man-in-the-middle attacks are possible. This problem is the same for CL-AKA. The ECQV implicit certificate is small and efficient, but some problems are apparent. First, unlike an explicit certificate that explicitly validates another certificate, a signature, and a message, the ECQV system verifies a certificate and a public key by calculating the public key from another certificate without verifying the transmitted message; there is no signature function. Replay and spoofing attacks are also possible. The sender requests an implicit certificate from the CA. As this is being transmitted for authentication, the attacker seizes and retransmits it, thus pretending to be a legitimate sender. Therefore, an implicit certificate should not be used alone; additional key agreement should be ensured using a key calculated from the implicit certificate.

Recently, many AKA schemes that use the ECQV to protect against KGC masquerade and key replace attacks have been proposed. In both 2015 [17] and 2017 [18], Sciancalepore et al. developed efficient, ECQV-based, implicit certificate-based AKA protocols for IoT environments. However, the work of [17] has a problem in that the session key generation information is exposed and there is no nonce in the message authentication code (MAC) value for the session information, so an external attacker could generate the session key by intercepting the transmitted data. Using this session key, masquerade attack was also possible. The work of [18] solved the problem of generating a session key, as described above, but there was a problem that the transmitted data could be retransmitted and used.

In addition to this, many AKA schemes that are performed quickly are proposed, including [19,20,21,22,23,24]. Abdmeziem et al. [19] propose an end-to-end key management protocol for e-health applications. The authors of [19] propose a protocol to ensure secure communication between constrained and unconstrained nodes using third parties. However, if a third party has malicious intent, there is a possibility of the session key being stolen and session hijacking through a man-in-the-middle attack.

Challa et al. [20] proposed an AKA scheme for cloud-assisted cyber-physical system (CPS) in 2018. The authors of [20] proposed a secure protocol for CPS environments such as smart grid through a trust authority, but a user masquerade attack is possible because the cloud server does not check the validity of the authentication request. In addition, there is a problem that communication can be performed without generating a session key.

Wazid et al. proposed [21,22] in 2018. The authors of [21] propose an authentication and key management protocol for a generic IoT network. A process in which a user performs a sensing node and lightweight AKA through a smartcard is proposed. They [21] do not use public keys, only exclusive or (XOR) operation, hash operation, and symmetric key encryption/decryption. Therefore, compared with the public key schemes, it is more efficient in terms of operation time, but only provides safety depending on the symmetric key. In addition, there is the problem that a malicious intermediate object can pretend to be a node and a user.

The work of [22] proposes an authentication and key management protocol for a cloud-assisted body area sensor network. A process in which a user executes a personal server and lightweight AKA through a mobile device is proposed. As in [21], it is a scheme that uses only XOR, hash operation, and symmetric key encryption/decryption, but it is an improved scheme by reducing unnecessary communication processes in intermediate objects. However, because the public key is not used, non-repudiation is not provided, and safety of parameters needs to be considered. Until recently, Wazid proposed authentication and key management schemes for various environments such as cloud-based IoT [23], fog computing services [24], Internet of Drones (IoD) [25], and implantable medical devices deployment [26]. The basics of these schemes are the similar to [21,22], and satisfy the security requirements in a specific environment.

#### 2.4.2. CL-AKA Schemes

Generally, CL-AKA schemes feature the following six algorithms. The Set-Secret-Value, Set-Private-Key, and Set-Public-Key algorithms are employed by the user to set the secret and public key pair. In the key agreement phase, users A and B create a common session key required for encryption via message exchange.

Setup: the KGC generates a public parameter and a master secret key (security parameter inputs).Partial-Key-Extract: the KGC generates a user’s partial private and public keys using the public parameter, master secret key, and user’s ID, and delivers it to the user.Set-Secret-Value: the user creates secret information by inputting the public parameter and his/her ID.Set-Private-Key: the user sets a private key by inputting the public parameter, partial private key, and secret information.Set-Public-Key: the user sets a public key by inputting the public parameter, his/her partial public key, and secret information.Key Agreement: users A and B generate messages using their IDs, public keys, and temporary keys. After exchanging messages, a common session key is generated using secret information. If the protocol is successful, the session keys generated by the two communicators will be identical.

CL-PKC-based authentication key agreement protocols without pairing were developed by Geng et al. [27] and Hou et al. [28] to increase computational efficiency. However, the public key used for AKA cannot be confirmed to be the public key of the sender, as described above. Efforts have been made to resolve this problem. Since then, many CL-AKA schemes have been proposed [29,30,31,32,33,34,35,36,37]. There are two common security requirements for CL-PKC technologies. First, because the public key and identifier must be verified without a certificate, a replacement attack on the public key is possible, unlike the existing PKI-based cryptographic technology. This is an attack performed by an attacker by replacing the user’s public key with a value generated by the attacker, and occurs because there is no certificate that serves as a signature for the public key. In addition, in CL-PKC, KGC generates partial secret keys to users, and attacks performed using partial secret keys should be considered. Therefore, CL-AKA should also consider public key replacement attacks and partial private key attacks by malicious KGC.

Yang et al. [29] proposed a certificateless key exchange scheme that does not use pairing operation in 2011. Although we propose AKA based on Diffie–Hellman key exchange between users who generated ID-based partial private keys through KGC, they are vulnerable to public key replacement attacks. The attacker can perform the authentication and key agreement process by being disguised as a legitimate participant through public key replacement, and can also generate the session key.

Kim et al. [30] proposed efficient CL-AKA between two objects in 2013. However, it is possible to perform a masquerade attack by retransmitting a value for generating session key as public key replacement attack.

Farouk et al. [31] proposes a two-party CL-AKA for the grid computing environment, but masquerade attack is also possible through public key replacement attack, and there is the problem that an attacker can legitimately generate a key. In addition, Xie et al. [32] and Park et al. [33] proposed pairing-free CL-AKA in 2016, but both schemes can perform masquerade attacks through public key replacement.

In Sun et al. [34] and Simplicio Jr et al. [35], because the partial key generated by KGC contains only the information of the user’s identifier and the verification tag, only the identifier can actually be checked. Later, Xie et al. [36] and Daniel et al. [37] solve this problem by including the information of verification public key generated by the user and the identifier in the partial key generated by KGC. However, public key replacement attacks are also possible in [36]. The authors of [37] argued that there was no problem even if the partial key generated by KGC was transmitted publicly. However, if a partial key is transmitted publicly, anyone can create values that are used as input when performing hash operation on the value to be authenticated. Therefore, it is necessary to consider parameter safety.

## 3. Security Requirements

### 3.1. Mutual Authentication

The most important security aspect of an IoT environment is authentication. Mutual authentication is essential during communications among multiple entities; key agreement is required.

### 3.2. Prevent Key Leakage

Authentication generates a session key for later use, and this must not be leaked. If an attacker derives or steals a key, all transmitted data will be exposed. Therefore, the key must not be leaked as a result of a public key replacement attack or replay attack.

### 3.3. Prevent Replay and Masquerade Attacks

Key leak is possible if a key is calculated via a transmitted message or retransmitted. The person who retransmits may be disguised as a legitimate user. A spoofing attack compromises availability to legitimate users; a party views the attacker as legitimate.

## 4. Proposed Schemes

We develop two schemes allowing two objects to communicate directly when establishing AKA in IoT environments. Existing ECQV-based key management protocols are at risk of node spoofing caused by replay attacks. To solve this problem, Scheme 1 eliminates unnecessary key generation processes and employs legitimate parameters. Scheme 2 is based on CL-AKA and proposes a way to explicitly verify the identifier and public key. In this section, firstly, the proposed model is explained, and the protocol for the two proposed schemes is explained in detail.

### 4.1. Proposed Model

The target model in this paper is an IoT service environment, and end-to-end authentication and key agreement between two objects constituting the IoT environment can be applied. For example, in the smart factory environment shown in Figure 1, IoT sensor devices located on the production line must communicate in real time. If an external attacker device participates in the production line network, it can transmit false information to the MES, causing financial and physical damage to the factory. Therefore, production line devices must exchange data with each other in real time, and Scheme 1 can be applied to environments that require such fast AKA. In addition, AKA is also required when transmitting data collected by sensor devices to the MES or when commanding devices from the MES. In particular, if the MES issues a command, sending an incorrect command message can cause great damage as well. In this situation, more reliable communication than Scheme 1 is required, and Scheme 2 can be applied. Figure 4 shows a model in which Schemes 1 and 2 can be applied in a smart factory environment. In the existing CL-PKC, a partial key was created through KGC, which generates a key, but it is unified and used as a CA to perform the roles of both Schemes 1 and 2. CA manages ECQV implicit certificate and partial key.

### 4.2. AKA via an ECQV Implicit Certificate (Scheme 1)

In this section, we propose Scheme 1 using ECQV so that two objects can communicate directly on authentication and key agreement in the IoT environment. In the existing ECQV-based key management protocol, node masquerade due to replay attacks has been a problem. To solve this problem, we propose an AKA protocol that reduces unnecessary processes in the key generation process and uses legitimate parameters. Figure 5 shows the scenario of Scheme 1 and the system parameters of Scheme 1 are as follows.

*: A communication participant (CA: certificate authority, A: device A, B: device B).$ID_{*}$: Identifier of the entity;$PU_{*},\;PR_{*}$: The public and private key pair of the entity;E: An elliptic curve on group G of prime order q;P: The generator on cyclic group G used to calculate the certificate;$\left(C_{*},\;\gamma_{*}\right)$: The ECQV implicit certificate of the entity;H(·): The cryptographic hash function;DS: The shared secret value used by the two objects to agree on the session key;KDF: The key derivation function;SK: The agreed key to be used in the current session.

#### 4.2.1. Setup Phase

In the setup phase, the devices participating in the IoT are registered in the CA. An ECQV implicit certificate is issued via registration. Thereafter, in the AKA phase, entities with implicit certificates can authenticate and negotiate keys without the intervention of the CA. Below, A sets up the issue of an implicit certificate $\left(C_{A},\gamma_{A}\right)$.

Step 1. A selects a random positive integer $k_{A}$ and generates a public elliptic curve point $R_{A}\;=\;k_{A}\cdot P$ and sends it to the CA.

Step 2. The CA selects a random positive integer $k_{CA}$ and generates an implicit certificate $C_{A}$ and an implicit signature $\gamma_{A}$ (points on the elliptic curve), as follows, and sends them to A. The full implicit certificates are $\left(C_{A},\gamma_{A}\right)$ pairs:(1)$$C_{A}\;=\;R_{A}+k_{CA}\cdot P$$
(2)$$\gamma_{A}\;=\;PR_{CA}+k_{CA}\cdot H\left(P_{A},ID_{A}\right)$$

Step 3. A computes a private key $PR_{A}$ and a public key $PU_{A}$, as shown below. Thus, A can generate pairs of implicit certificates $\left(C_{A},\gamma_{A}\right)$ and public keys $\left(PR_{A},PU_{A}\right)$ through the CA. The implicit certificate is verified if the public key can be successfully restored from the implicit certificate because the content is calculated by the CA when generating the public key pair.
(3)$$PR_{A}\;=\;\gamma_{A}+k_{A}\cdot H\left(C_{A},ID_{A}\right)$$
(4)$$PU_{A}=PR_{A}\cdot P$$

#### 4.2.2. Authentication and Key Agreement Phase

The entities registered in the CA engage in mutual authentication using their implicit certificates for session key generation that guarantees secure communication; they agree on a session key. During this process, a key derivation function that prevents the possible replay and spoofing attacks to which conventional schemes are exposed is applied. Below, the AKA that allows A to communicate securely with B is described.

Step 1. A selects a random positive integer $r_{A}$ and sends an ECQV implicit certificate $\left(C_{A},\gamma_{A}\right)$ to B together with $r_{A}$ and a personal identifier.

Step 2. B restores the public key $PU_{A}$ using the implicit certificate and identifier received from A as follows. This confirms that A has been issued a certificate by the CA.
(5)$$PU_{A}=PU_{CA}+C_{A}\cdot H\left(C_{A},ID_{A}\right)$$

Step 3. B calculates the shared secret value *DS* to be used by A and B when generating a session key as follows:(6)$$DS=PR_{B}\cdot PU_{A}=PR_{B}\cdot PR_{A}\cdot P$$

Step 4. B selects any positive integer $r_{B}$. Using this, the $r_{A}$ and identifier received from A, as well as the *DS,* are inputs to the key derivation function *KDF*, which calculates $K_{DS}$ as follows:(7)$$K_{DS}=KDF\left(DS,ID_{A},ID_{B},r_{A},r_{B}\right)$$

Step 5. B sends $r_{B}$, its implicit certificate $\left(C_{B},\gamma_{B}\right),$ and an identifier $ID_{B}$ to A. Then, the session key $SK=H\left(K_{DS}\right)$ is calculated to prepare for secure communication with A.

Step 6. A restores the public key $PU_{B}$ as follows using its implicit certificate and the identifier received from B. This confirms that B has successfully received a certificate from the CA.
(8)$$PU_{B}=PU_{CA}+C_{B}\cdot H\left(C_{B},ID_{B}\right)$$

Step 7. A calculates the shared secret value *DS* to be used to generate the same session key as created by B as follows. *DS* can be computed by only A and B using the elliptic curve discrete logarithm approach.
(9)$$DS\;=\;PR_{A}\cdot PU_{B}\;=\;PR_{A}\cdot PR_{B}\cdot P$$

Step 8. A inputs the identifier $r_{B}$ received from B and the *DS* generated by itself into the key derivation function *KDF* to calculate a $K_{DS}$, which is the same as that generated by B, as follows:(10)$$K_{DS}\;=\;KDF\;=\;\left(DS,ID_{A},ID_{B},r_{A},r_{B}\right)$$

Step 9. A calculates the session key $SK\;=\;H\left(K_{DS}\right)$ using $K_{DS}$. Thereafter, an encrypted message can be transmitted to B (which prepared the secure communication). Interaction is now possible.

### 4.3. Proposed AKA with Pairing-Free Certificateless PKC (Scheme 2)

Scheme 2 (based on CL-AKA) confirms the existence of a public key in Scheme 1. Two objects can communicate directly during AKA in an IoT environment. Current ECQV-based key management protocols cannot confirm the existence of a public key. They are faster than AKA schemes based on PKI certificates, but their security strengths are lower. To solve this problem, we link the public key to the verification value. Scheme 2 verifies the user and public keys. Scheme 2 features the certificateless-based AKA introduced in 2.4; however, to bind the public key to the signature generated by the CA, the user first generates a key pair and a partial secret key in the CA. This is the scheme described previously [37]. Figure 6 shows the scenario of Scheme 2 and the system parameters of Scheme 2 are as follows.

*: A communication participant (CA: certificate authority, A: device A, B: device B).$ID_{*}$: The identifier of an entity;E: An elliptic curve on group G of prime order q;P: The generator of cyclic group G;s: The CA master secret key;$P_{pub}$: The CA master public key;$sv_{*},\;pv_{*}$: The verification private/public key pair generated by an entity;$D_{*}$: The partial key of an entity;$Pr_{*},Pu_{*}$: The full private and public key pair;$H_{1}\left(\cdot \right)$: The mapping hash function $H_{1}:{\left\{0,1\right\}}^{*}\times G^{2}\to Z_{q}^{*}$;$H_{2}\left(\cdot \right)$: The mapping hash function $H_{2}:{\left\{0,1\right\}}^{*}\times {\left\{0,1\right\}}^{*}\times {\left\{0,1\right\}}^{*}\times G^{4}\to Z_{q}^{*};$$H_{k}\left(\cdot \right)$: The one-way hash function;SK: The session key to be used (generated via agreement).

#### 4.3.1. Setup Phase

In the setup phase, the CA generates the initial parameters employing the Setup (*k*) algorithm that uses the security parameter *k*. The CA then obtains a master secret key *s* and generates a master public key $P_{pub}\;=\;s\cdot P$. The CA then creates public parameters and registers devices that request registration. The devices create individual public and private key pairs using the UserKeyGeneration (*params,*
$ID_{i}$) algorithm and send their identifiers and public keys to the CA for registration. The CA signs off on device requests via the ExtractPartialKey (*params, s,*
$ID_{i},\;pv_{i}$) algorithm and generates and returns partial secret keys (the $ppk_{i}$ values). Each device receiving a partial secret key generates a static private/public key pair using the SetPrivateKey (*params,*
$ID_{i},ppk_{i},sv_{i})$ and SetPublicKey (*params,*
$ID_{i},ppk_{i},pv_{i}$) algorithms.

Step 1. The CA selects a security parameter *k* and generates a master secret key *s*. Then, a master public key $P_{pub}\;=\;s\cdot P$ is generated, as is the public parameter *params*; both are released via the Setup (*k*) algorithm as follows:(11)$$params\;=\;\left\{G,q,P,P_{pub},H_{1},H_{2},H_{k}\right\}$$

Step 2. A device *i* that wishes to receive a partial secret key from the CA first generates an individual public/private key pair $pu_{i},\;sv_{i}$ employing the UserKeyGeneration (*params, ID*) algorithm. Device *i* chooses $x_{i}\in_{R}Z_{q}^{*}$ and calculates $pu_{i}\;=\;x_{i}\cdot P$ and $sv_{i}$ as $sv_{i}\;=\;x_{i}$.

Step 3. Device *i* sends its identifier $ID_{i}$ and public key $pu_{i}$ to the CA, which uses the ExtractPartialKey (*params, s,*
${\mathrm{ID}}_{i},\;pv_{i}$) algorithm to generate a partial secret key for that device. The CA selects $r_{i}\in_{R}Z_{q}^{*}$ and generates $R_{i\;}=\;r_{i}\cdot P$ and a signature $z_{i}\;=\;r_{i}+s\cdot H_{1}\left(ID_{i},pv_{i},R_{i}\right)$ for the public key. The CA then transmits the partial secret key $ppk_{i}\;=\;\left(R_{i},z_{i}\right)$ to device *i* via a secure channel.

Step 4. Device *i* receiving the partial secret key $ppk_{i}$ generates a personal static secret key $Pr_{i}$ employing SetPrivateKey (*params,*$ID_{i},ppk_{i},sv_{i}$) and sends a static public key $Pr_{i}$ via SetPublicKey (*params,*$ID_{i},ppk_{i},pv_{i}$). $Pu_{i}$ is generated as follows:(12)$$Pr_{i}\;=\;sv_{i}+z_{i}$$
(13)$$Pu_{i}\;=\;\left(pu_{i},R_{i},Z_{i}\;=\;z_{i}\cdot P\right)$$

#### 4.3.2. Authentication and Key Agreement Phase

When A, which has received a partial secret key from the CA, wishes to communicate securely with B via AKA, the KeyAgreement algorithm is enlivened.

Step 1. A selects an ephemeral secret key $t_{A}\in_{R}Z_{q}^{*}$ and computes an ephemeral public key $T_{A}\;=\;t_{A}\cdot P$.

Step 2. A sends ${\mathrm{ID}}_{B},ID_{A},Pu_{A}$, and $T_{A}$ to B.

Step 3. B verifies $T_{A}\in G$ for $\mathrm{the}\;T_{A}$ received from A and confirms that this is the public key of A generated by the CA, using the following formula:(14)$$Z_{A}?\;=\;R_{A}+P_{pub}\cdot H_{1}\left(ID_{A},pu_{A},R_{A}\right)$$

Step 4. When verification is complete, B also chooses an ephemeral secret key $t_{B}\in_{R}Z_{q}^{*}$ and computes an ephemeral public key $T_{B}=t_{B}\cdot P$.

Step 5. B sends ${\mathrm{ID}}_{A},ID_{B},Pu_{B}$, and $T_{B}$ to A.

Step 6. A verifies $T_{B}\in G$ for ${\mathrm{the}\;T}_{B}$ received from B and confirms that it is the public key of B generated by the CA using the following Equation:(15)$$Z_{B}?\;=\;R_{B}+P_{pub}\cdot H_{1}\left(ID_{B},pu_{B},R_{B}\right)$$

Step 7. If verification is confirmed, A calculates $S_{B}\;=\;pu_{B}+Z_{B}$, and B calculates $S_{A}\;=\;pu_{A}+Z_{A}$. Both A and B generate the session information data e and d as follows:(16)$$e\;=\;H_{2}\left(ID_{A},ID_{B},S_{A},S_{B},T_{A},T_{B}\right)$$
(17)$$d\;=\;H_{2}\left(ID_{B},ID_{A},S_{B},S_{A},T_{B},T_{A}\right)$$

Step 8. A creates $\sigma_{\mathrm{AB}}$ and B creates $\sigma_{\mathrm{BA}}$ as follows:(18)$$\sigma_{AB}\;=\;\left(dt_{A}+Pr_{A}\right)\cdot \left(eT_{B}+S_{B}\right)$$
(19)$$\sigma_{BA}\;=\;\left(et_{B}+Pr_{B}\right)\cdot \left(dT_{A}+S_{A}\right)$$

The equivalence of $\sigma_{\mathrm{AB}}$ and $\sigma_{\mathrm{BA}}$ above is verified as follows in $S_{i}\;=\;x_{i}P+z_{i}P\;=\;\left(x_{i}+z_{i}\right)\cdot P\;=\;Pr_{i}\cdot P$:(20)$$\sigma_{AB}\;=\;\left(dt_{A}+Pr_{A}\right)\cdot \left(eT_{B}+S_{B}\right)\;=dt_{A}\cdot eT_{B}+dt_{A}\cdot S_{B}+Pr_{A}\cdot eT_{B}+Pr_{A}\cdot S_{B}\;=det_{A}t_{B}P+dt_{A}Pr_{B}P+et_{B}Pr_{A}P+Pr_{A}Pr_{B}P\;=\left(et_{B}+Pr_{B}\right)\cdot \left(dT_{A}+S_{A}\right)=\sigma_{BA}$$

Thereafter, $H\left(\sigma_{AB}\right)$ and $H\left(\sigma_{BA}\right)$ are calculated using $\sigma_{AB},\;\sigma_{BA}$ and used as session keys.

## 5. Analysis of Proposed Schemes

We now compare and analyze the two schemes and show that they meet the security requirements set out in Section 3.

### 5.1. Mutual Authentication

As mentioned above, the most important security factor in an IoT environment is authentication. Mutual authentication is essential to guarantee secure communication, as is key agreement. Our two schemes are key agreement protocols, and implicit certificates are issued via ECQV, as in previous works. If an entity’s public key can be restored using an implicit certificate, the user is implicitly authenticated. In the real world, a replay or a spoof attack, or a key leak, may occur.

In Scheme 1, the elliptic curve Diffie–Hellman (ECDH) algorithm is applied using a public key restored by an implicit certificate, solving certain problems. Equation (6) shows that the DS is generated via an ECDH-based key agreement that only A and B can calculate. It is possible to generate a $K_{DS}$ that in turn generates a session key (via the key derivation function KDF) by inputting $ID_{A},\;ID_{B}$ and $r_{A},r_{B}$; these are the identifiers and random positive integers (nonces) used to create the DS and establish the session. The only entities that can calculate these are A and B, and mutual authentication is assured via calculation of $K_{DS}$.

Scheme 2 performs authentication using partial keys based on the Schnorr signature. Each partial key is generated by the CA and is verified using both the CA and object’s public key. If A first sends a public key $Pu_{A}$ and tag $T_{A}$ to B, the validity of A’s public key can be checked using Equation (14). Similarly, if B sends a public key $Pu_{B}$ and tag $T_{B}$ to A, the validity can be checked employing Equation (15). Mutual authentication is performed in this manner.

### 5.2. Prevent Key Leakage

In existing schemes [17,18,29,31,36], an attacker can generate a key by simply eavesdropping on transmitted data. To solve this problem, the key is generated during authentication and the key agreement phase in the session. An arbitrary value was used during key agreement, but key leakage via a replay attack was not prevented. In our two present schemes, key leakage is possible if anyone other than A and B can generate a session key. Therefore, we make it impossible to derive a key via messages transmitted over a public channel.

In Scheme 1, calculation of $PU_{A}$ (the public key of A; Equation (5)) can be performed by an attacker. However, Equation (6), which calculates the secret value *DS* for calculation of the session key, can be performed only using the private key of B. An attacker seeking the session key $SK=H\left(K_{DS}\right)$ faces considerable difficulty, equivalent to that experienced when seeking to solve the elliptic curve discrete logarithm problem (ECDLP) of $PR_{B}\cdot PU_{A}=PR_{B}\cdot PR_{A}\cdot P$.

In Scheme 2, the public keys A and B received from the CA can be validated using Equations (14) and (15). A’s public key $Pu_{A}$ is composed of $pu_{A},R_{A},\mathrm{and}\;Z_{A}$. Even if the attacker knows the public key and calculates e,d using public information, it is difficult to calculate a session key $\sigma_{AB}$ or $\sigma_{BA}$ because the attacker lacks the secret keys A and B. In other words, it is difficult for an attacker to obtain a session key; the difficulty is identical to that experienced when seeking to solve the ECDLP of $\left(pu_{A}+Z_{A}\right)=\left(sv_{A}+z_{A}\right)\cdot P$ to determine the secret key A.

### 5.3. Prevent Replay and Masquerade Attacks

Most existing schemes use ECDH for key agreement. The attacker participates in communication using the certificate and the arbitrary value transmitted from a sender to a receiver; both replay and spoof attacks are possible. Existing schemes aim to participate in communication by disguising as a legitimate user through key theft, replay, and public key replacement attacks. Therefore, if a security threat occurs in the existing schemes, it becomes a cause of masquerade attack. In our schemes, keys are generated by adding identifiers of A and B; these are not available to anyone who seeks to attack using $\mathrm{the}\;r_{A},r_{B}$, and $K_{DS}$ generated by the user.

### 5.4. Efficiency

The simulation environment featured an Intel i5-4690 3.50-GHz CPU processor, 16 GB RAM, and the Windows 10 operating system. In both schemes, the Koblitz elliptic curve $y^{2}=x^{3}+ax+b\;\left(mod\;p\right)$, where a=1 and b is a 163 bit random prime defined on $F_{2^{163}}$, was used to provide safety equivalent to that of a 1024 bit RSA.

In Scheme 1, we reduced the number of communication times compared with the existing [17,18,19,20,21,22] scheme, which enabled us to speed up the process of authentication and key agreement. Figure 7 compares the times required for AKA by the ECDSA with existing schemes and our schemes (which are faster). The proposed scheme using ECQV is faster than using general ECDSA, and the speed of the proposed scheme is slightly faster. Table 3 shows the comparison of Scheme 1 with the existing schemes.

In Scheme 2, the computational overhead is better than that of an existing CL-AKA scheme. Compared with Scheme 1, which can perform authentication and key agreement quickly, it does not show efficiency in terms of speed. However, compared with the existing ECDSA or other schemes [29,30,32,33,34,35,36,37], the computation speed of Scheme 1 is reduced, and it provides safety considering masquerade attack, replay attack, and public key verifiability. Table 4 shows the comparison of Scheme 2 with the existing schemes.

## 6. Conclusions and Future Research

In increasingly large IoT environments, AKA is essential for secure communication. AKA protocols have been investigated for many years, and efforts have been made to render them lightweight for IoT. In particular, in the IoT environment, various cryptographic technologies such as authentication, authorization, and access control are being studied according to the service environment and security requirements. In recent years, data generated by IoT devices are stored and managed through cloud/fog computing [38]. However, several security requirements existing schemes must be satisfied. We present two AKA protocols to provide end-to-end security in IoT environments such as a smart factory. Our two schemes are mutually authenticated via an implicit certificate or public key issued by the CA.

Scheme 1 uses ECQV implicit certificates for authentication. Scheme 2 derives partial keys using CL-PKC. ECQV implicit certificate enables fast and efficient authentication, but public keys can be attacked. The public key lacks a signature; a receiver performing authentication cannot know whether it is the public key of a sender or a “fallback” attacker. Scheme 2 secures the public key via Schnorr signature, employing CL-PKC to verify the key, but is slower than Scheme 1. Both schemes resolve current security problems and meet the security requirements of Section 3. The existing schemes are open to replay and public key replacement attacks and key leakage. We thus minimized transmitted data and increased the communication speed.

We only describe AKA for an end-to-end IoT environment in this paper. However, the components of the IoT environment are diverse and the network structure can also change. Rather than the end-to-end structure between a single device and server, it can be 1: N communication between multiple devices and servers, and this structure can be hierarchical. In the future, we will evaluate AKA techniques that can solve problems arising when gateway objects are used in hierarchical/multiple sources rather than end-to-end communication IoT environments. In the former environments, the devices are very heterogeneous, as are the network configurations. In addition, research on authorization and access control technologies closely related to authentication in a cloud/fog computing environment is also required as the recent IoT service changes, and should be lighter than existing schemes.

## Figures and Tables

**Figure 1 sensors-20-05350-f001:**
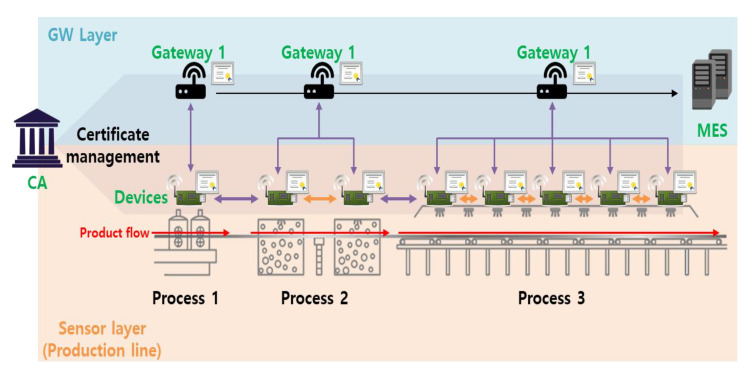
Example model: smart factory environments. MES, manufacturing execution system; CA, certificate authority.

**Figure 2 sensors-20-05350-f002:**
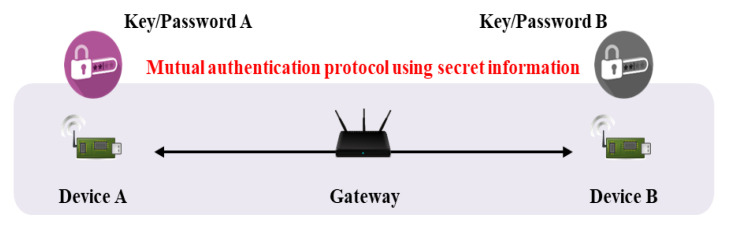
Mutual authentication flow with gateway.

**Figure 3 sensors-20-05350-f003:**
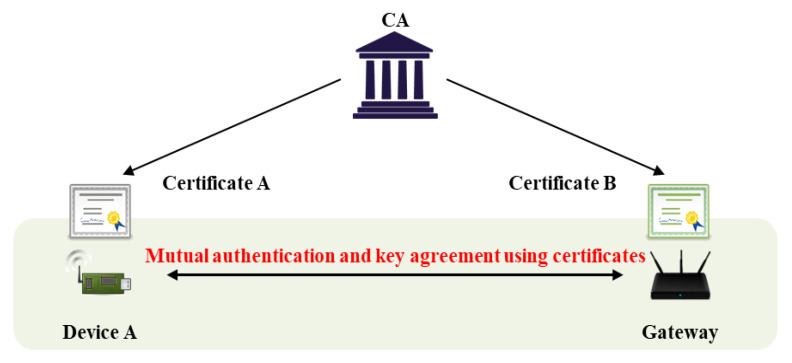
Mutual authentication flow with certificate.

**Figure 4 sensors-20-05350-f004:**
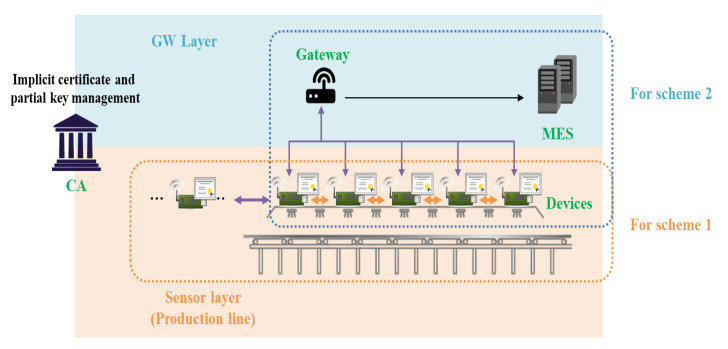
Proposed Internet of Things (IoT) smart factory service model for Schemes 1 and 2.

**Figure 5 sensors-20-05350-f005:**
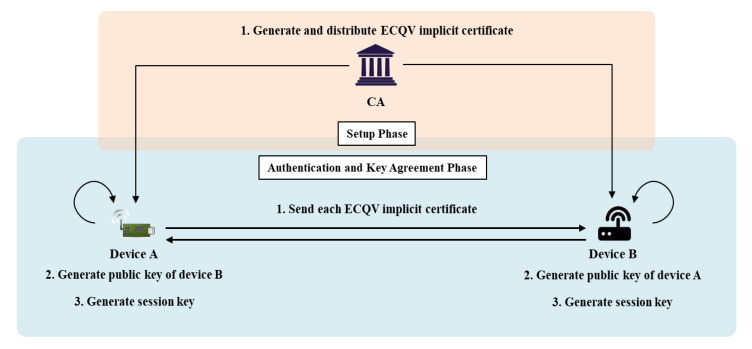
The scenario of Scheme 1. ECQV, elliptic curve Qu–Vanstone.

**Figure 6 sensors-20-05350-f006:**
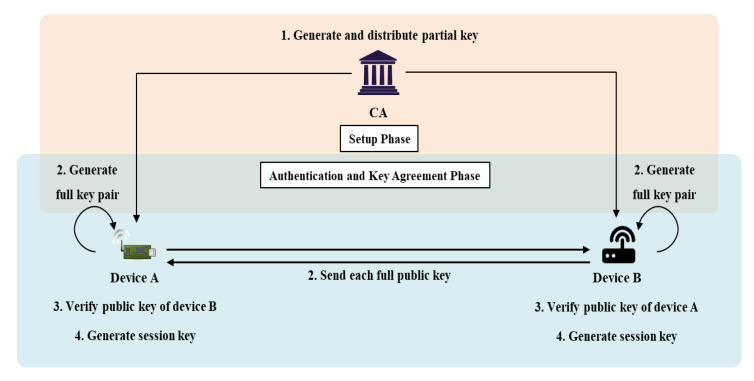
The scenario of Scheme 2.

**Figure 7 sensors-20-05350-f007:**
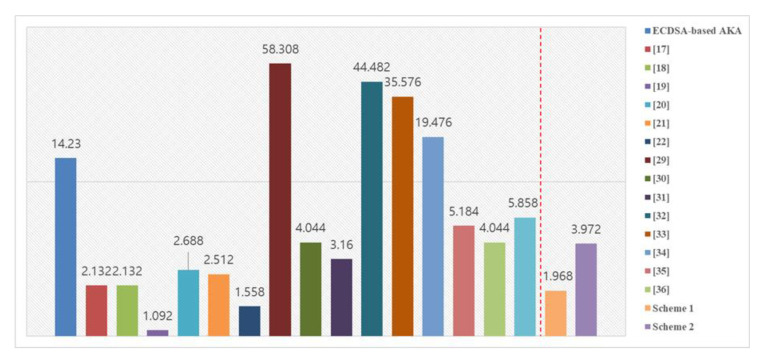
Comparison of key agreement time with existing schemes.

**Table 1 sensors-20-05350-t001:** A comparison of explicit and implicit certificate. PKI, public key infrastructure; ECC, elliptic curve cryptography.

	Explicit Certificate	Implicit Certificate
**Key Derivation**	Included in the certificate	Must be calculated using certificates and signatures
**Public Key Verification**	Signature verification using a public key	No verification process
**Structure**	Identifier, public key, electronic signature	Identifier, public key recovery data
**Comparison**	The key and certificate sizes are relatively large (PKI); slow	The key and certificate sizes are relatively small (ECC); fast

**Table 2 sensors-20-05350-t002:** A comparison of security strengths. ECQV, elliptic curve Qu–Vanstone.

Strength	Key Length (Bits)		Certificate Size (Bits)		
	ECC	RSA	ECQV	ECDSA	RSA
80	192	1024	193	577	2048
112	224	2048	225	673	4096
128	256	3072	257	769	6144
192	384	7680	358	1153	15,360
256	521	15,360	522	1564	30,720

**Table 3 sensors-20-05350-t003:** Comparison of Scheme 1.

	[17]	[18]	[19]	[20]	[21]	[22]	**Proposed Scheme 1**
**Prevent Key Leakage Attack**	O	O	X Possible to steal key during connection	O	X The key of the node is leaked to the user	O	O
**Prevent Masquerade**	X Possible to masquerade as a result of replay	X Possible to masquerade as a result of replay	X Possible session hijacking during connection	X Possible to masquerade without key agreement	X Possible to masquerade via node’s key	X Cannot provide non-repudiation of sender	O
**Prevent Replay Attack**	X No nonce in MAC	X Reuse of published values	O	X Reuse of published values	O	O	O
**Public Key Verifiability**	∆ Only implicit authentication	∆ Only implicit authentication	∆ Only implicit authentication	X Only use pre-shared symmetric key	X Only use pre-shared symmetric key	X Only use pre-shared value	∆ Only implicit authentication
**Operation**	2EA + 4EM + 4h	2EA + 4EM + 4h	4SE + 6h	4EM + 22h	8SE + 16h	19h	2EA + 4EM + 2h

O (X): scheme is strong (weak) in this category, ∆: scheme is partial strong in this category, EA: elliptic curve addition operation, EM: elliptic curve scalar multiple operation, SE: symmetric key encryption, h: one-way hash function.

**Table 4 sensors-20-05350-t004:** Comparison of Scheme 2.

	[29]	[30]	[31]	[32]	[33]	[34]	[35]	[36]	[37]	Proposed Scheme 2
**Prevent Key Leakage Attack**	X Possible to leakage as a result of replay	O	X Possible to leakage as a result of masquerade	O	O	O	O	X Possible to replay with leaked temporary key	O	O
**Prevent Masquerade**	X Possible to masquerade by replay	X Possible to masquerade by replay	X Possible to masquerade by key replace	X Possible key replaces and masquerade	X Possible key replaces and masquerade	X Possible key replaces and masquerade	X Cannot verify public key	X Possible to masquerade by replay	X Possible to masquerade by replay	O
**Prevent Replay Attack**	X Possible to replay	X Possible to replay	X Possible to join session with masquerade	O	O	X Possible man-in-the-middle attack	O	O	O	O
**Public Key Verifiability**	X Possible to public key replacement	O	O	X Cannot verify public key	X Cannot verify public key	X Cannot verify public key	X Cannot verify public key	O	X Cannot verify public key	O
**Operation**	12E + 11EM + 7h	10EA + 8EM + 4h	10EA + 6EM + 4h	2P + 8EA + 8EM + 4h	8E + 4h	36EA + 40EM + 14h	6EA + 10EM + 8h	10EA + 8EM + 4h	8EA + 12EM + 5h	6EA + 8EM + 4h

O (X): scheme is strong (weak) in this category, E: exponential operation, P: pairing operation, EA: elliptic curve addition operation, EM: elliptic curve scalar multiple operation, h: one-way hash function.

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
