# Peer review of "A Lightweight Authentication and Key Agreement Schemes for IoT Environments"

_sensors, 2020, doi:10.3390/s20185350_

Round 1

Reviewer 1 Report

The contributions seem well presented along with reference scenario, security requirements elicitation, proposed conceptual scheme, experiment setup and analysis of results. However, the related works are not well covered.

-       The authentication and access control (i.e., authorization) mechanisms are two interrelated issues. The security policies can be used as an authentication mechanism. The author can have a look existing/relevant dynamic access control mechanisms, ref.: the dynamic CAAC (context-aware access control) models from this survey - A survey of context-aware  access  control  mechanisms  for  cloud and fog networks:  Taxonomy and open research issues: Sensors 2020, and write a future research direction here.

-       The authors also can have a look at the direction of ‘accessing data from multiple IoT sources’ and ‘providing a secure authentication mechanism’ in such direction'. The author at least can provide a future research direction focusing multiple IoT sources/environments. The last section can be named as 'Conclusion and Future Research'. Ref: IoT Streaming Data Integration from Multiple Sources, Computing journal, 2020.

-       How to bring data from multiple IoT sources and protect them from unauthorized users and cyber attackers/criminals? – this can be explained and included in the related work section. In the literature, the ''dynamic access control study'' has been appeared that are relevant to this work: A secure lightweight three-factor authentication scheme for IoT in cloud computing environment, Sensors, 2019. The authors also can explore access control models/protocols; and how the access control protocols can be incorporated as cybersecurity mechanism. The secure data access/authorization concept should be incorporated along with secure Authentication challenges.

-       The authors should explore the authorization-related works from the literature. The authors should cover a good literature survey in this direction and provide a comparative assessment table, covering both authentication and authorization domains – highlighting the contributions in this paper comparing with other works. Recently, scholars are trying to introduce dynamic access control approaches/frameworks when data are coming from distributed/multiple cloud environments – here both authentication and authorization need to be considered as security mechanism. There is also a very recent domain of fog computing and edge-oriented security to safeguard IoT data at the edge of the network. 

Author Response

Dear Reviewer,

Sincerely,

Dae-Hwi Lee.

Reviewer 2 Report

The authors in this paper are trying to propose a lightweight authentication scheme as well as key agreement model for IoT environments. IoT is weak devices when it comes to computation and calculation! The proposed scheme suppose to improve the authentication process. The authors must handle the following: - The smart factory environments is not explained in the results section? - The topology of the system is missing. Adding figure will help. - Simulation environment table with full details? - Comparison with other similar models also missing! Fig. 6 is not enough. - Why page 11 is half empty! - Enrich your paper and include more studies from recent years with consideration to blockchain. I recommend: A Blockchain-Based Decentralized Composition Solution for IoT Services, Trustworthy and Sustainable Smart City Services at the Edge, Efficient and Robust Top-k Algorithms for Big Data IoT, Empowering reinforcement learning on big sensed data for intrusion detection. - Proofread the whole paper.

Author Response

Dear Reviewer,

Sincerely,

Dae-Hwi Lee.

Reviewer 3 Report

I think overall the content is there the writing however is weak. English quality should be significantly improved in the revision.

References and related work should be strengthened

The following key references are missing

Vajda I, Buttyán L. Lightweight authentication protocols for low-cost RFID tags. InSecond Workshop on Security in Ubiquitous Computing–Ubicomp 2003 Oct 12 (Vol. 2003). Gilbert H, Robshaw M, Sibert H. Active attack against HB/sup+: a provably secure lightweight authentication protocol. Electronics Letters. 2005 Oct 24;41(21):1169-70. Malina L, et al.. A secure publish/subscribe protocol for internet of things. InProceedings of the 14th International Conference on Availability, Reliability and Security 2019 Aug 26 (pp. 1-10).   Kamal M, et al. Blockchain-Based Lightweight and Secured V2V Communication in the Internet of Vehicles. IEEE Transactions on Intelligent Transportation Systems. 2020 Jun 24.   More emphasis should be given on the Results section    

Author Response

Dear Reviewer,

Sincerely,

Dae-Hwi Lee.

Round 2

Reviewer 2 Report

The author answered are minimal effort. They are advised to revise and spend more effort convincing the reader and the reviewer as the current version does not constitute enough contributions.

Author Response

Dear Reviewer,

Thank you for your careful reading.

Sincerely,

Dae-Hwi Lee.

Reviewer 3 Report

The authors have handled all of my revision requests from the first round. Paper should undergo a thorough spelling and grammar check prior to final publication

Author Response

(The authors gave the same response as above.)
